# Thermoelectric Properties of Nickel and Selenium Co-Doped Tetrahedrite

**DOI:** 10.3390/ma16030898

**Published:** 2023-01-17

**Authors:** Duarte Moço, José F. Malta, Luís F. Santos, Elsa B. Lopes, António P. Gonçalves

**Affiliations:** 1C2TN, DECN, Instituto Superior Técnico, Universidade de Lisboa, Campus Tecnológico e Nuclear, 2695-066 Bobadela, Portugal; 2Centre for Physics of the University of Coimbra (CFisUC), Department of Physics, University of Coimbra, 3004-516 Coimbra, Portugal; 3CQE, Departamento de Engenharia Química, Instituto Superior Técnico, Universidade de Lisboa, 1049-001 Lisboa, Portugal

**Keywords:** tetrahedrite, thermoelectric power factor, isovalent doping, electronic properties simulation

## Abstract

As the search continues for novel, cheaper, more sustainable, and environmentally friendly thermoelectric materials in order to expand the range of applications of thermoelectric devices, the tetrahedrite mineral (Cu_12_Sb_4_S_13_) stands out as a potential candidate due to its high abundance, low toxicity, and good thermoelectric performance. Unfortunately, as most current thermoelectric materials achieve zTs above 1.0, ternary tetrahedrite is not a suitable alternative. Still, improvement of its thermoelectric performance has been achieved to zTs ≈ 1 via isovalent doping and composition tuning, but most studies were limited to a single doping element. This project explores the effects of simultaneous doping with nickel and selenium in the thermoelectric properties of tetrahedrite. Simulated properties for different stoichiometric contents of these dopants, as well as the measured thermoelectric properties of the correspondent materials, are reported. One of the samples, Cu_11.5_Ni_0.5_Sb_4_S_12.5_Se_0.5_, stands out with a high power factor = 1279.99 µW/m·K^2^ at 300 K. After estimating the thermal conductivity, a zT = 0.325 at 300 K was obtained for this composition, which is the highest for tetrahedrites for this temperature. However, analysis of the weighted mobility shows the presence of detrimental factors, such as grain boundaries, disorder, or ionized impurity scattering, pointing to the possibility of further improvements.

## 1. Introduction

It is estimated that, from the total primary energy globally produced from all sources, only 34% provides useful work. Of the unused 66% part, the largest portion, accounting for 54% of the overall primary energy, is lost as heat. As a result, from sustainability, ecological, and economic points of view, it is imperative to develop cheap and easy to implement technologies that can harness the lost energy, specifically the waste heat, and improve the global energy efficiency [1,2,3].

A technology that allows a direct conversion of heat into electricity was developed just after the World War II, with the settlement of the first thermoelectric generators, in the form of a soviet partisan mess kit radio that could be powered by a small cooking fire. Over the following decades, other thermoelectric systems were developed to reliably produce energy for extended periods of time without maintenance or any additional human intervention, finding applications in remote automated equipment, such as lighthouses, polar expedition devices, space probes, extraterrestrial rovers, or a reliable power source for pacemakers. Unfortunately, despite the great potential of this technology, it was founded that there was little use outside of these niche applications, as other methods proved to be more efficient and cost-effective. However, recent developments in materials sciences, predictive computational modules, as well as advances in micro/nanofabrication, reignited the interest in this field of study, with new and more efficient thermoelectric materials being discovered every year [4,5,6,7,8,9,10,11].

The operating principle of a thermoelectric generator lies in the Seebeck effect, a phenomenon first discovered in 1821 by the German physicist Thomas Joan Seebeck. The Seebeck effect consists in the development of an electrical potential difference when a temperature gradient is applied to a material, as charge carriers energized by higher temperatures in the warmer region move to the colder region, resulting in a higher concentration of electrons [4]. To make proper use of this effect in a thermoelectric generator, two types of legs, usually n-type and p-type semiconductors, are assembled alternatingly in a series, as represented in Figure 1 [12,13]. N-type materials are responsible to move negative charge carriers, such as electrons, from the warmer region to the cold one, while the p-type materials are responsible for transporting positive quasiparticles called “holes” that behave as charge carriers. The movement of such positive quasiparticles from the warmer to the cold region is equivalent to a movement of negative charge carriers from the cold to the warm region, which allows the current flow when the n-type and p-type materials are assembled in a series [14,15]. Therefore, pairs of n-type and p-type semiconductors with good thermoelectric properties are needed to settle efficient thermoelectric generators.

The performance of a material for thermoelectricity can be evaluated by its thermoelectric dimensionless figure of merit,
(1)$$\mathrm{zT}=\frac{S^{2}\sigma }{\kappa }$$
which depends on the electrical conductivity, σ, thermal conductivity, κ, and Seebeck coefficient, S. To maximize zT, it is necessary to have a high electrical conductivity to better move the charges across the legs, a high Seebeck coefficient to have a high electrical potential difference, and a low thermal conductivity to maintain the temperature gradient (the driving force behind the movement of charge carriers). As these properties are deeply interconnected, it is difficult to adjust each property individually and obtain a material with optimum thermoelectric performance. Conceptually, a thermoelectric generator with an average zT ≈ 1.0 in the thermoelectric materials can achieve a maximum 10% conversion efficiency with a 250 K temperature difference, with an average zT ≈ 2.0, to achieve the same 10% conversion efficiency only a 150 K temperature difference is needed [16,17,18,19].

Currently, there are already several thermoelectric materials with high thermoelectric performances (zTs above 1.0) for application in a wide range of temperature intervals. However, the thermoelectric materials currently used in commercial devices are usually composed of rare and/or toxic elements, such as lead, bismuth, tellurium, etc., which makes them too expensive and too hazardous (to the human health and environment) to be used in most circumstances. Thus, the scientific community is trying to develop novel, cheaper, and less toxic thermoelectric materials, without compromising their thermoelectric performance [19,20,21,22,23,24].

One promising candidate among the novel thermoelectric materials is tetrahedrite, Cu_12_Sb_4_S_13_, a naturally occurring earth-abundant mineral composed of significantly fewer toxic elements, that has good thermoelectric properties as a p-type semiconductor (zT ≈ 0.6 at 673 K). This performance accounts from the unusual low thermal conductivity that stems from its large and complex crystal structure. Tetrahedrite has a cubic crystal structure ($I\bar{4}3m$ space group), composed of 58 atoms per unit cell, with the elements distributed into five distinct crystallographic sites. Copper atoms have two different chemical environments: Cu1, located in a tetragonal configuration with three S atoms and one Sb atom; and Cu2, a near coplanar triangle configuration with three S atoms. The S atoms are also distributed into two distinct chemical environments: S1, with 12 out of 13 S atoms, arranged in a tetragonal configuration; and S2, the remaining S atom, surrounded by six Cu atoms in octahedron configuration. All four Sb atoms have the same chemical environment, with a lone electron pair and bonding to three S atoms, forming an SbS_3_ trigonal pyramid. The tetrahedrite complex structure has two main peculiarities: (i) the Cu2 atoms are able to strongly vibrate around their equilibrium position, displaying large anisotropic atomic thermal displacement parameters; and (ii) due to the lone pair in the Sb atoms, the tetrahedral environment of Sb is incomplete, without a fourth bond. These peculiarities contribute to a reduced lattice contribution to the thermal conductivity, as the interaction between them results in the modification of the Cu2 environment into an oversized atomic cage, effectively functioning as an inharmonic rattler scattering phonon center, which, according to the phonon glass/electron crystal principle, leads to low thermal conductivities [25,26,27,28,29,30,31,32,33,34,35].

Unfortunately, tetrahedrite in its ternary composition has a zT ≈ 0.6 at 673 K, which cannot compete with the current thermoelectric materials that have a zT around or higher than 1.0. However, extensive studies have shown that this value can be improved via chemical substitution, or “doping”, by replacing Cu with a wide range of transition metals, Sb with As, Bi, and Te, and S with Se [26,32]. While many studies have been conducted on the effects of a single dopant, the study of the impact of two or more simultaneous dopants is still a relatively unexplored field of research. The present work seeks to determine the effect of doping with Ni and Se on the tetrahedrite thermoelectric properties, with the scope of improving the overall thermoelectric performance [25,36].

## 2. Materials and Methods

The synthesis process was based on previous studies [37,38,39], mainly following the methodology employed by Alves, et al. [40]. Cu_12−x_Ni_x_Sb_4_S_13−y_Se_y_ (0 ≤ x ≤ 1.5; 0 ≤ y ≤ 1.5) samples were synthesized by reacting pure elements, Cu (99.9999%), Ni (99.9999%), Sb (99.999%), S (99.9999%), and Se (99.99%) from Alfa Aesar and Sigma Aldrich, inside quartz ampoules sealed under vacuum (10^−1^ Torr). The content of each element was calculated for the desired stoichiometric ratio to make a 1.5 g mixture. To offset sulfur losses due to evaporation, an additional ~1.0 wt% sulfur excess was added. The samples were submitted to a multiple step process: (I) first, the samples were heated up to 1173 K at the rate of 4 K/min and kept at that temperature for 1 h; (II) then, they were cooled to 973 K at the rate of 11 K/min and kept for 10 min; after which (III) they were cooled to 923 K at the rate of 11 K/min and kept for 20 min; and finally, (IV) they were cooled to 673 K the rate of 4 K/min and kept for 18 h before being removed.

After casting, the samples were submitted to an annealing step, starting by being crushed into a fine powder in a mortar and then shaped into pellets by cold pressing with a hydraulic press. The disks were subsequently sealed under vacuum (10^−1^ Torr) inside quartz ampoules and subjected to a temperature treatment at 723 K for 7 days. Undoped tetrahedrite samples and samples doped only with Se underwent an annealing process at 573 K for 14 days and at 623 K for 14 days, respectively, due to higher tendency to produce secondary phases.

The characterization of the samples was carried out via powder X-ray diffraction (XRD), Raman spectroscopy, and scanning electron microscopy coupled with energy dispersive spectroscopy (SEM/EDS). XRD was performed on D2 Phaser Bruker 2nd Gen, by measuring in the 10–65° range with a 0.02° step and taking 0.6 s per step, through 1.0 mm slit. The PowderCell package, Diffrac.EVA, Origin 90E and UnitCell software [41] were used to analyze the data and calculate the lattice parameter of the samples. Raman spectroscopy measurements were carried out with the LabRAM HR Evolution Raman spectrometer with 532 nm laser. A portion of the samples (after casting and after annealing) was manually polished with SiC paper and analyzed by SEM/EDS with a JEOL JSM-7001F, field emission gun scattering electron microscope, Tokyo, Japan, equipped for EDS with an Oxford Instruments light elements detector, High Wycombe Buckinghamshire, UK.

The measurement of electrical resistivity and Seebeck coefficient was made in the 20–350 K temperature range by employing a modified version of the method used by Chaikin and Kwak [42]. Two small pieces were removed from the annealed samples and shaped to a needle-like piece (2 × 0.5 × 0.5 mm^3^), for the Seebeck coefficient measurements, and a cuboid/flat prismatic shaped piece (2 × 1.5 × 0.5 mm^3^), for the electrical resistivity measurements. The resistivity was measured with the standard four-probe method, using silver paint electrical contacts. As for the Seebeck effect, the piece was glued to two sheets of gold, which are glued to two bars of quartz connected to gold probes coated with silver paint.

Electronic band structure calculations and thermoelectric properties simulations were performed based on the method employed by Ravaji et al. [43] and Knízek et al. [44], which combines the WIEN2k software [45], for band calculations and density of states simulations, with BoltzTraP software [46], for the simulation of the temperature dependence of the Seebeck coefficient, electrical conductivity, and thermal conductivity.

WIEN2k package software solves Kohn–Sham equations in the density function theory (DFT) with the linear augmented plane wave (LAPW) method. The onsite Coulomb repulsion U combined with generalized gradient approximation (GGA + U) was implemented for the presented calculations. To improve the description of 3d electrons, the parameters U = 4 eV and J = 1 eV were used in such method (U_eff_ = 3 eV). 

The calculations of the thermoelectric properties, Seebeck coefficient (*S*), electrical resistivity (ρ = 1/σ), and the thermal conductivity (κ = κ_L_ + κ_e_), were performed by applying the Boltzmann transport theory through the BoltzTrap package with a constant relaxation time approximation for charge carriers. To simplify the calculations, it was assumed that κ_L_ = 0.5 W K^−1^ m^−1^, as reported by Suekuni et al. [34] for samples doped with Ni. These properties can be calculated with the formulas:(2)$$\sigma_{\mathrm{ij}}=K_{\mathrm{ij}}^{0}$$
(3)$$S_{\mathrm{ij}}=\frac{K_{\alpha j}^{1}/K_{\alpha i}^{0}}{\mathrm{eT}}$$
(4)$$K_{\mathrm{ij}}^{e}=\frac{K_{\mathrm{ij}}^{2}}{e^{2}T}$$
(5)$$K_{\alpha \beta }^{n}\left(T,\;\mu \right)=\int \frac{{\tau e}^{2}.\partial \epsilon .\partial \epsilon }{\hbar^{2}\partial k_{\alpha }\partial k_{\beta }}{\left(\epsilon -\mu \right)}^{n}\left[-\frac{\partial f_{\mu }\left(T,\epsilon \right)}{\partial \epsilon }\right]d\epsilon$$
where ϵ, is the energy variable, T the absolute temperature, µ the chemical potential, *e* is the electron charge, and τ the relaxation time. The main weight in the abovementioned integration has partially occupied states over the characteristic energy µ ± 3 kT, which is approximately ±75 meV around the chemical potential at 300 K.

## 3. Results

### 3.1. Electronic Band Structure Calculations

The density of states of some of the compositions, calculated with spin polarization, are presented in Figure 2. The ternary tetrahedrite (Cu_12_Sb_4_S_13_) converged in a non-magnetic solution, and since the Fermi level resides below the top of the valence band complex, it can be described as an almost metallic heavily doped p-type semiconductor. After doping with Ni, the band gap reduces, and tetrahedrite becomes increasingly more conductive, also displaying a ferromagnetic behavior. The valence band moves above the Fermi level with spins down, as Ni is a known ferromagnetic material, which explains why the spin asymmetry increases with Ni doping. Additionally, since Ni^2+^ has an electronic configuration of [Ar]3d8, which contrasts to Cu^2+^ with an electronic configuration of [Ar]3d9, doping results in one less electron, or, in other words, in one more “hole”, that increases the number of states available below the Fermi level.

### 3.2. Thermoelectric Properties Simulations

The thermoelectric properties simulated with Boltztrap are shown in Figure 3. The electrical resistivity increases with the increase in temperature for all compositions, which points to a metallic behavior rather than to a semiconductor behavior. Another indication of metallic behavior is the Seebeck coefficient decrease with the temperature decrease. Since the Seebeck coefficient is always positive, it can be concluded that “holes” are the main charge carriers. Thus, the metallic behavior obtained from simulations point to tetrahedrite as a highly degenerate p-type semiconductor.

Regarding the effect of the dopants on the thermoelectric properties, simulations show that the electrical resistivity increases after doping both with Ni and with Se. Taking a closer look at the zT’s behavior with changes in composition, simulations suggest that doping with Ni and Se, aside from the Cu_12_Sb_4_S_12_Se composition, will result in thermoelectric efficiencies higher than in the undoped tetrahedrite, with the highest value, achieving an estimated zT ≈ 0.30 at 300 K, being observed for the Cu_11.5_Ni_0.5_Sb_4_S_12.5_Se_0.5_ composition. These results, while promising, must be taken with care, as various assumptions were considered in the calculations, including the assumption that lattice contributions to the thermal conductivity do not exceed κ_L_ = 0.5 W K^−1^ m^−1^.

Contrary to what was expected from other experimental studies and band calculations [37,39,47], Se doping resulted in a resistivity higher than the ternary tetrahedrite. However, a particularly low resistivity, only slightly higher than the ternary tetrahedrite, is calculated for the Cu_12_Sb_4_S_12_Se composition.

### 3.3. X-Ray Diffraction and Raman Spectroscopy

Powder X-ray diffractograms of as-cast and annealed samples are shown in Figure 4. In all samples, the most intense peaks were identified as belonging to tetrahedrite, but some peaks that do not match to this phase are also present and were identified as famatinite (Cu_3_SbS_4_) and covellite (CuS). After annealing, the peaks of secondary phases become less intense and frequent, with some samples just exhibiting the tetrahedrite diffractogram.

Raman spectra of both as-cast and annealed samples are shown in Figure 5. All samples exhibit the anomalies or peaks typical of the tetrahedrite phase (at 315 cm^−1^ and 348 cm^−1^), with an occasional observation of peaks that have been identified as belonging to anilite (Cu_7_S_4_). Additionally, other peaks related to the Cu-S and Sb-S bonds can be also observed, but instead of representing secondary phases, they are connected to the material ejected from the samples due to the high intensity of the laser beam used in Raman spectroscopy analysis. Apart from a slight shift of the 348 cm^−1^ peak (up to 355 cm^−1^) related to energy deviations of the molecular vibrations induced by substitutions, no noteworthy changes in the Raman spectra between as-cast and annealed samples were observed.

The unit cell lattice parameter of the tetrahedrite phases, calculated from the XRD data using the Unit Cell software, are presented in Figure 6. Overall, the lattice parameter ranges between 10.322–10.369 Å in as-cast samples and 10.321–10.382 Å in annealed samples. The lattice parameter increases with the increase in Se concentration, with the samples containing Se presenting higher lattice parameters after annealing. By contrast, Ni introduction in tetrahedrite does not appear to have a noticeable effect on the lattice parameter.

### 3.4. SEM/EDSs

The SEM observation of Cu_12−x_Ni_x_Sb_4_S_13−y_Se_y_ samples often indicates the presence of other phases in addition to the matrix: a darker one, frequently with a star shape and composed of copper and/or nickel sulfides, (Cu,Ni)_x_S_y_; and an intergranular lighter phase, mainly composed of chalcostibite-pribramite, CuSb(S,Se)_2_. The number and size of observed phases significantly decrease after annealing, with some compositions being monophasic. A SEM image of a typical Cu_12−x_Ni_x_Sb_4_S_13−y_Se_y_ sample is presented in Figure 7.

Semi-quantitative analysis of the matrix phase composition of each sample was obtained through SEM/EDS and is summarized in Table 1 and Table 2 for as-cast and annealed samples, respectively. 

The EDS analysis of the matrix composition of as-cast samples reveals a tetrahedrite phase, being in good agreement with the envisaged (nominal) composition but normally exhibiting amounts of Cu slightly higher than expected. Ni and Se are detected in the matrix of the samples containing them, which confirms the integration of both dopants on the tetrahedrite phase. The presence of covellite and other copper sulphides is frequently observed. After annealing, the composition slightly changes and become closer to the nominal composition, indicating a higher integration of Ni and Se in the tetrahedrite phase, but usually the Se content is lower than expected, with an average deficit of approximately 17.3%. Being semi-quantitative, the results shown in Table 1 and Table 2 should be regarded as indicative.

### 3.5. Thermoelectric Properties

The results of the thermoelectric properties measurements are presented in Figure 8. The electrical resistivity increases as the temperature decreases for all samples, which is characteristic of semiconductor behavior. The Seebeck coefficient also decreases as the temperature decreases, indicating a metallic behavior, which points to tetrahedrite as highly degenerate semiconductors, in agreement with the simulations. The Seebeck coefficient is always positive pointing to “holes” as the main charge carriers, indicating tetrahedrite as a p-type semiconductor.

In general, the higher the Ni concentration the higher the electrical resistivity and Seebeck coefficient, which is in agreement with previous studies [34,48]. Regarding the Se content, it was frequently not possible to obtain pure tetrahedrite materials from solely Se-doped samples. However, the effect of Se doping in the Seebeck coefficient seems to be smaller than the effect on electrical resistivity (varying between 77 and 83 µV/K at 300 K), and a linear relation appears to exist between the Seebeck coefficient and the Se-content.

## 4. Discussion

The WIEN2k simulations show a density of states characteristic of a p-type semiconductor for all tetrahedrites, regardless of the type and content of the dopant, with the Fermi level located just below the border between the valence band and a small gap (≈1.2 eV). This small overlap points to the metallic behavior that tetrahedrite exhibits experimentally, further indicating that this material, in spite of the amount of Ni or Se doping (within the studied limits), behaves as a highly degenerate p-type semiconductor. The simulations also show that doping with Ni and Se results in a thermoelectric performance higher than in the ternary tetrahedrite, with the exception of the tetrahedrite with a concentration of Se y = 1.0. It also points out the Cu_11.5_Ni_0.5_Sb_4_S_12.5_Se_0.5_ tetrahedrite as the one with the highest performance, presenting a zT around 0.30 at ~300 K. It should be mentioned that, although the current simulations present results similar to other studies, there is a significant difference: the resulting DOS (Figure 2) indicates that the ternary tetrahedrite and tetrahedrite doped only with Se have a small spin asymmetry in the density of states, which is not evident in the other studies and suggests a magnetic ground state. Furthermore, the simulation of thermoelectric properties of the ternary tetrahedrite does not show the characteristic peak related to the metal-semiconductor transition at 85 K, indicating that the current model has some limitations. Nonetheless, it was possible to predict what are the compositions that present the best performance, which was confirmed by the experimental results.

All samples have tetrahedrite as the major phase, even just after casting. However, XRD, Raman, and SEM/EDS analyses indicate that some samples still have secondary phases after this step, namely chalcostibite (CuSbS_2_), famatinite (Cu_3_SbS_4_), covellite (CuS), and anilite (Cu_7_S_4_), which tend to disappear when annealed. In as-cast samples, the most commonly observed secondary phases were the copper–nickel sulphides, followed by CuSb(S,Se)_2_. Aside from the Cu_10.5_Ni_1.5_Sb_4_S_11.5_Se_1.5_ sample, the others only exhibited one of these secondary phases. A potential explanation for the appearance of both phases in Cu_10.5_Ni_1.5_Sb_4_S_11.5_Se_1.5_ might be the high dopant concentration, which is only worsened by the apparent excess of Se measured in the matrix (EDS revealed a particularly high Se content of 2.00, rather than the expected 1.5). It should be mentioned that in about one third of the samples it is possible to discern two different tetrahedrite phases with some small differences in lattice parameters and composition. With the annealing step, the secondary phases tend to disappear and only one tetrahedrite phase is observed. Pores also became smaller, and the EDS analysis of the matrix showed a composition closer to the nominal one. An increase in the tetrahedrite lattice parameters with the rise of Se content is observed in as-cast samples, which most probably results from the ionic radius difference between Se^2-^ (1.98 Å) and S^2-^ (1.84 Å). A further rise after annealing is observed in the Se-doped samples, indicating an increase in the Se concentration in the tetrahedrite matrix, which was confirmed by SEM/EDS analysis. No significant changes were observed in the lattice parameters of Ni-doped samples, as the difference between the ionic radius of Ni^2+^ (0.69 Å) and Cu^2+^ (0.73 Å) is small. However, SEM/EDS analysis confirms the presence of Ni in the tetrahedrite phase. To summarize, annealing is a fundamental step to reduce the concentration of secondary phases and homogenize the tetrahedrite phase.

The thermoelectric properties of the samples only containing Ni are identical to those reported in other studies, showing a similar temperature dependence of electrical resistivity and Seebeck coefficient, which increase with the increase in Ni concentration [34,48]. From the band structure calculations, doping with Ni is expected to increase the hole concentration and, consequently, the electrical conductivity, but aspects, such as grain boundary electron or impurity scattering can strongly influence the electrical conductivity. However, the observed behavior was explained as caused by a shift of the Fermi level to the top of the valence band that results in a decrease in the carrier concentration. Moreover, the overall Seebeck coefficient values of the present samples are slightly lower and the electrical resistivity slightly higher than what was reported [21,34,37,47,49,50,51,52]. 

Regarding the Se-doped materials, previous studies showed a decrease in the electrical resistivity with the increase in Se concentration, reaching a minimum for y = 1.0 and increasing for higher Se contents (up to y = 2.0) but remaining lower than the undoped materials [37]. However, in the present work, it was not possible to correctly compare the results, as samples frequently showed secondary phases, in particular due to the tendency to degenerate into the CuSb(S,Se)_2_ phase, even after adjusting the annealing conditions. The sole exception was the Cu_12_Sb_4_S_11.5_Se_1.5_ sample that, while maintaining the main tetrahedrite phase after annealing, is significantly more resistive than the undoped tetrahedrite, contrary to what was expected from other studies [37,39]. This difference is most likely related to the lack of the shaping step (SPS or hot-press), as the reported studies were made on materials submitted to it, while the present results come from only annealed samples. Thus, the samples investigated in this work are expected to have higher porosity and more secondary phases, cracks, and other structural or microstructural inhomogeneities than those analyzed in the other studies, which have a detrimental effect on the thermoelectric properties.

The weighted mobility (electron mobility weighted by the density of electronic states) is an important parameter to characterize semiconducting materials [53]. In the free electron model, the weighted mobility is a (temperature dependent) material property independent of doping that can be calculated using the measured electrical resistivity and Seebeck coefficient:(6)$$\mu_{w}=\frac{331}{\rho }{\left(\frac{T}{300}\right)}^{-3/2}\left[\frac{\exp\left[\frac{\left|S\right|}{k_{B}/e}-2\right]}{1+\exp\left[-5\left(\frac{\left|S\right|}{k_{B}/e}-1\right)\right]}+\frac{\frac{3}{\pi^{2}}\frac{\left|S\right|}{k_{B}/e}}{1+\exp\left[5\left(\frac{\left|S\right|}{k_{B}/e}-1\right)\right]}\right]$$
with k_B_ representing the Boltzmann constant and *e* the electronic charge. In most good thermoelectric materials, the weighted mobility decreases as temperature increases following a T(^−3/2^) progression, which is a hint of phonon scattering. However, in other systems, grain boundaries, disorder, or ionized impurity scattering can also contribute to it and lead to a decrease with the decreasing temperature. For example, the grain size can be small enough to produce high resistances and affect negatively the charge mobility leading to an increase in the weighted mobility with the increase in temperature and a deviation from the T^−3/2^ progression [53].

The weighted mobility as a function of temperature for the Cu_12−x_Ni_x_Sb_4_S_13−y_Se_y_ samples studied in this work is presented in Figure 9. Several conclusions can be taken from this data: (i) the values of the weighted mobility are small when compared with other good thermoelectric materials [53]; (ii) the progression of weighted mobility with temperature is sample dependent; (iii) the highest weighted mobility values are generally achieved when the Ni or Se stoichiometry is around 0.5–1.0.

The good thermoelectric properties of tetrahedrites mainly derive from their extremely low thermal conductivity [32]. The reduced contribution of electrical transport properties to the good thermoelectric performance of tetrahedrites was previously evidenced by the low power factors they exhibit, which was now confirmed by the small, weighted mobility values. However, most likely it will be possible to significantly improve such properties, as the different weighted mobility progressions with temperature observed in the present work point to a detrimental effect of factors, such as grain boundaries, disorder, or ionized impurity scattering. The best material seems to be the tetrahedrite with the Cu_11.5_Ni_0.5_Sb_4_S_12.5_Se_0.5_ composition, which exhibits the highest weighted mobility values and a T^−3/2^ progression. Nevertheless, a more systematic study of the tetrahedrites weighted mobility is needed and is under way.

The PF at 300 K as a function of composition is presented in Figure 10A. Overall, the present results show that doping with both Ni and Se can improve the thermoelectric properties of tetrahedrite, generally resulting in higher PFs. 

The highest PF at 300 K was achieved by the sample with the Cu_11.5_Ni_0.5_Sb_4_S_12.5_Se_0.5_ composition, which is in line with what was predicted by the simulations and exhibited by the weighted mobility behavior. A maximum PF of 1279.99 µW/m.K^2^ at 300 K is observed for this sample, followed by the Cu_11_NiSb_4_S_12.5_Se_0.5_ and Cu_11.5_Ni_0.5_Sb_4_S_12_Se compositions, with a PF = 527.96 µW/m.K^2^ and PF = 335.56 µW/m.K^2^ at 300 K, respectively.

Although the thermal conductivity (κ_T_) was not measured in this work, an estimation of the electronic contribution for thermal conductivity, κ_e_, can be made through the Wiedmann–Franz law,
(7)$$L=\frac{\kappa_{e}}{\sigma T}$$

By assuming an average lattice contribution for thermal conductivity of κ_L_ = 0.5 W K^−1^ m^−1^ in tetrahedrites [16,54], it is possible to estimate the total thermal conductivity, κ_T_, by applying the equation:(8)$$\kappa_{T}=\kappa_{L}+\kappa_{E}=\kappa_{L}+\left(L*T\right)/\rho$$
where L represents the Lorenz number, calculated through the formula L (10^−8^ V^2^/K^2^) = 1.5 + exp(-|*S*|/116) [55,56]. With this approximation, a figure of merit zT = 0.325 was calculated at 300 K for the sample with Cu_11.5_Ni_0.5_Sb_4_S_12.5_Se_0.5_ composition (Figure 10B), which is a particularly high value when compared to the reported results for tetrahedrites [34,37,49,57] . Other high zTs were also achieved with close compositions, zT = 0.207 for Cu_11_NiSb_4_S_12.5_Se_0.5_ and zT = 0.118 for Cu_11.5_Ni_0.5_Sb_4_S_12_Se. Since the best results are located around the region of x and y values between 0.5 and 1.0 in agreement with the simulation results, it can be concluded that the Wien2K-BoltzTrap method is an effective tool in predicting the thermoelectric properties of tetrahedrites.

## 5. Conclusions

The present study indicates that, while the synthesis methodology can produce samples mostly composed of tetrahedrite, it is still lacking in producing high-quality samples. Other steps must be added to the preparation procedure, namely a shaping (SPS or hot-press) step. The tuning of the casting and/or annealing conditions may also be successfully used to improve the thermoelectric performance of the samples, since it is necessary to decrease grain boundaries, disorder, or impurities that have a detrimental effect on the PF.

Despite the non-optimal synthesis procedure, the electronic band structure calculations and thermoelectric properties simulations indicate that a simultaneous doping with Ni and Se can significantly improve the thermoelectric performance of tetrahedrite, which was experimentally confirmed, culminating in the Cu_11.5_Ni_0.5_Sb_4_S_12.5_Se_0.5_ optimum composition, with a PF of 1279.99 µW/m.K^2^ and an estimated zT of 0.30 at 300 K. The weighted mobility analysis proved to be a good tool to identify the presence of factors that can have a detrimental effect on the thermoelectric properties and a way to help the identification of the optimal synthesis procedure. Therefore, a systematic study of the weighted mobility of simultaneous Ni and Se doped tetrahedrites is now underway.

## Figures and Tables

**Figure 1 materials-16-00898-f001:**
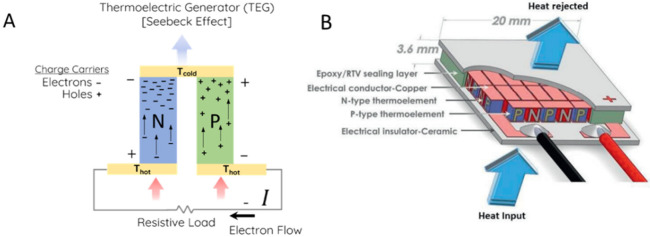
Representation of a thermoelectric generator: (**A**) Flow of charge carriers in a pair of n- and p-type leg pair; (**B**) schematic of thermoelectric generator module [12,13].

**Figure 2 materials-16-00898-f002:**
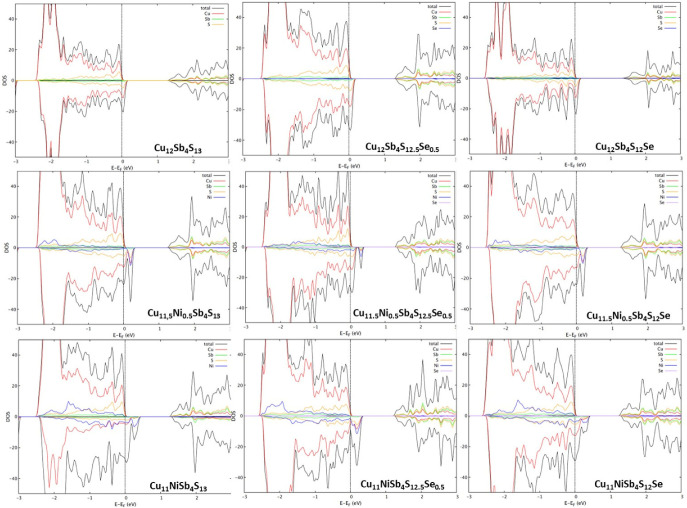
Density of states calculated through the WIEN2k package for tetrahedrite with specific stoichiometric content of Ni and Se, in accordance with the formula Cu_12−x_Ni_x_Sb_4_S_13−y_Se_y_ (0 ≤ x ≤ 1.0; 0 ≤ y ≤ 1.0).

**Figure 3 materials-16-00898-f003:**
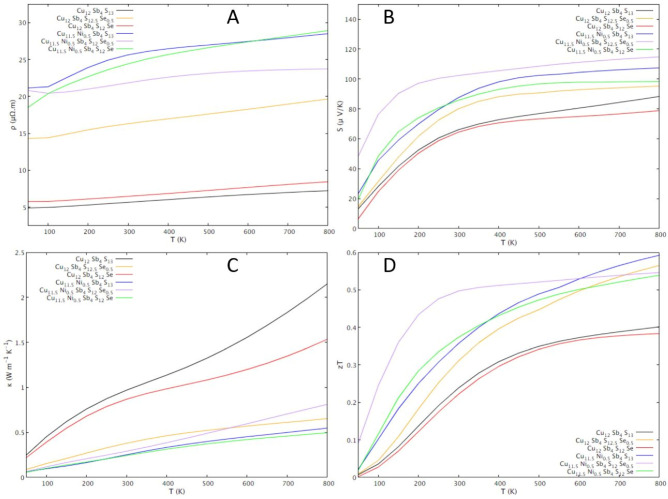
BoltzTrap simulations of the temperature dependence of thermoelectric properties of tetrahedrite with different dopant contents following the formula Cu_12−x_Ni_x_Sb4S_13−y_Se_y_ (0 ≤ x ≤ 1.0; 0 ≤ y ≤ 1.0): (**A**) Seebeck coefficient; (**B**) electrical resistivity; (**C**) thermal conductivity; (**D**) figure of merit, zT.

**Figure 4 materials-16-00898-f004:**
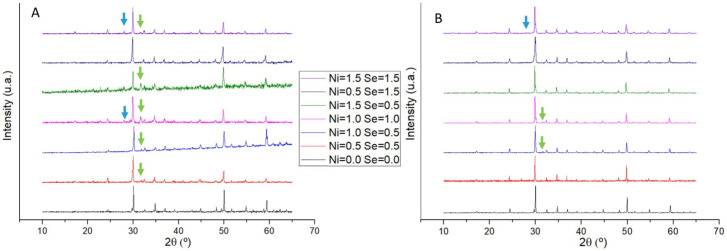
Powder X-ray diffractograms of Cu_12−x_Ni_x_Sb_4_S_13−y_Se_y_ samples ((**A**) as-cast and (**B**) annealed). Arrows indicate peaks of secondary phases: green for covellite (CuS) and blue for famatinite (Cu_3_SbS_4_).

**Figure 5 materials-16-00898-f005:**
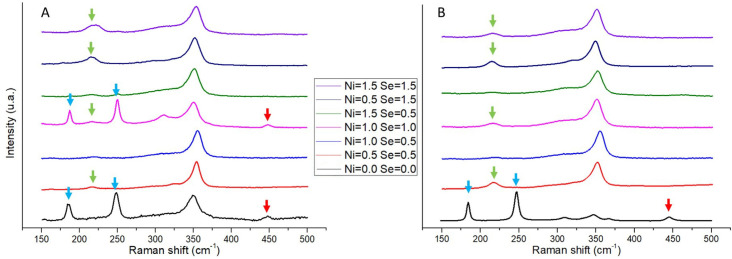
Raman spectra of Cu_12−x_Ni_x_Sb_4_S_13−y_Se_y_ samples after casting (**A**) and after annealing (**B**). The peaks pertaining to the tetrahedrite phase are located in the 315 and 348 cm^−1^ shifts. Arrows indicate the peaks not related to the tetrahedrite phase: green arrows—the Cu-S chemical bond of ejected material; blue arrows—the Sb-S chemical bond of ejected material; and red arrows—the Anilite (Cu_7_S_4_) secondary phase.

**Figure 6 materials-16-00898-f006:**
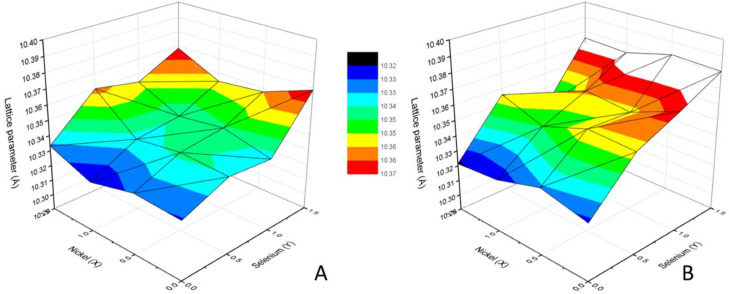
Lattice parameter dependence on Ni content (x) and Se content (y) in as-cast (**A**) and annealed (**B**) Cu_12−x_Ni_x_Sb_4_S_13−y_Se_y_ samples.

**Figure 7 materials-16-00898-f007:**
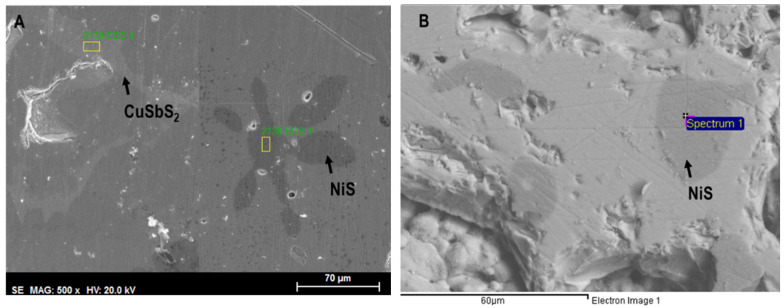
SEM backscattered images of (**A**) as-cast Cu_10.5_Ni_1.5_Sb_4_S_11.5_Se_1.5_ sample, with the lighter toned CuSb(S,Se)_2_ phase and the dark (Cu,Ni)_x_S_y_ stars; (**B**) annealed Cu_10.5_Ni_1.5_Sb_4_S_11.5_Se_1.5_ sample with a dark toned (Cu,Ni)_x_S_y_ phase.

**Figure 8 materials-16-00898-f008:**
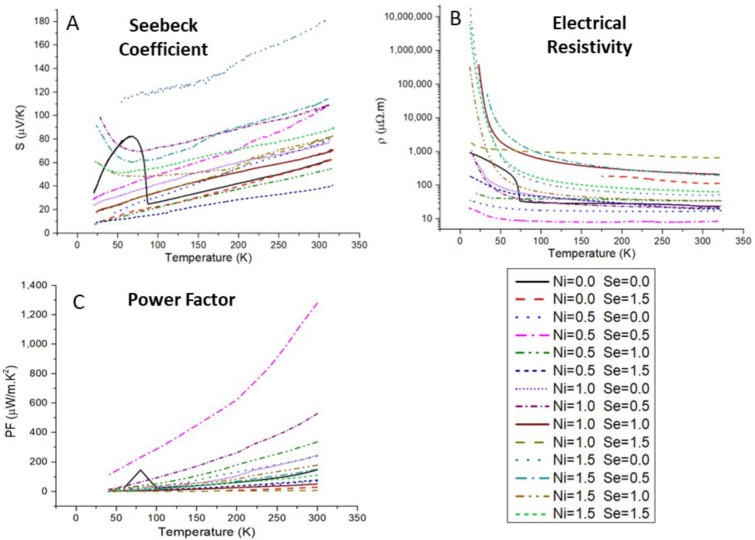
Measured temperature dependence of electrical resistivity, ρ (**A**) and Seebeck coefficient, *S* (**B**); and resulting power factor, PF, as a function of temperature (**C**), calculated for the Cu_12−x_Ni_x_Sb_4_S_13−y_Se_y_ samples.

**Figure 9 materials-16-00898-f009:**
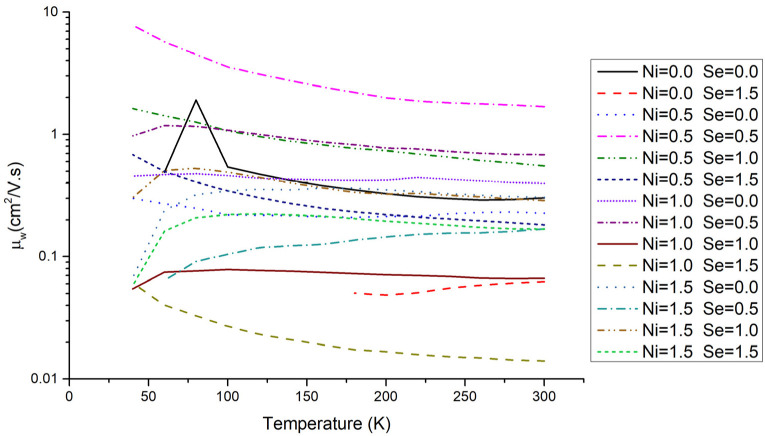
Weighted mobility as a function of temperature for the Cu_12−x_Ni_x_Sb_4_S_13−y_Se_y_ annealed samples.

**Figure 10 materials-16-00898-f010:**
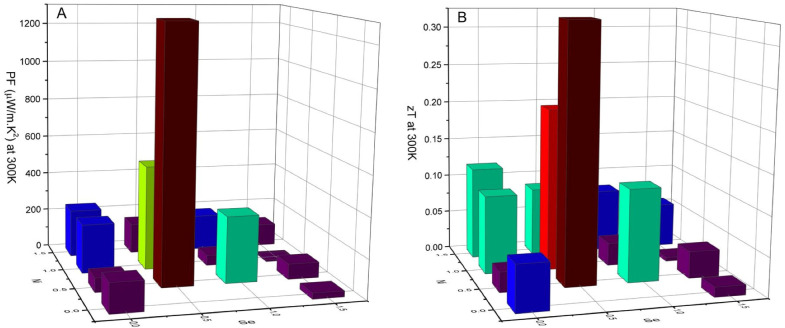
Power factor, PF, (**A**) and estimated figure of merit, zT, (**B**) at 300 K as a function of composition for the Cu_12−x_Ni_x_Sb_4_S_13−y_Se_y_ annealed samples.

**Table 1 materials-16-00898-t001:** Tetrahedrite EDS semi-quantitative analysis on Cu_12−x_Ni_x_Sb_4_S_13−y_Se_y_ as-cast samples.

Ni Content ^1^	Se Content ^1^	Cu (at%)	Ni (at%)	Sb (at%)	S (at%)	Se (at%)	Molecular Formula ^2^
0	0	48.1	0	12.9	39.0	0	Cu_13.94_Sb_3.75_S_11.31_
0.5	0	44.4	2.1	12.1	41.3	0	Cu_12.88_Ni_0.62_Sb_3.52_S_11.97_
0	0.5	41.3	0	14.4	43.4	0.9	Cu_11.98_Sb_4.18_S_12.58_Se_0.26_
0.5	0.5	39.6	1.2	13.0	44.7	1.5	Cu_11.50_Ni_0.35_Sb_3.76_S_12.96_Se_0.43_
1	0	47.4	4.2	13.2	35.2	0	Cu_13.74_Ni_1.23_Sb_3.84_S_10.20_
1	0.5	43.5	4.2	12.8	38.2	1.3	Cu_12.62_Ni_1.22_Sb_3.71_S_11.08_Se_0.37_
0	1	47.3	0	12.6	35.6	4.6	Cu_13.71_Sb_3.95_S_10.31_Se_1.32_
0.5	1	47.0	1.9	13.2	34.8	3.2	Cu_13.64_Ni_0.54_Sb_3.83_S_10.08_Se_0.92_
1	1	46.8	1.6	14.0	35.4	2.3	Cu_13.57_Ni_0.46_Sb_4.06_S_10.26_Se_0.66_
1.5	0	44.6	5.4	12.5	37.5	0	Cu_12.94_Ni_1.56_Sb_3.63_S_10.88_
1.5	0.5	44.1	4.8	12.8	38.3	0	Cu_12.80_Ni_1.38_Sb_3.70_S_11.12_
1.5	1	44.2	4.5	12.4	33.9	5.0	Cu_12.82_Ni_1.31_Sb_3.59_S_9.82_Se_1.46_
0	1.5	44.2	0	13.3	35.8	6.8	Cu_12.82_Sb_3.85_S_10.37_Se_1.96_
0.5	1.5	42.0	1.2	12.1	38.3	6.4	Cu_12.17_Ni_0.35_Sb_3.50_S_11.12_Se_1.86_
1	1.5	41.2	3.3	11.9	40.0	3.7	Cu_11.94_Ni_0.94_Sb_3.45_S_11.59_Se_1.08_
1.5	1.5	44.2	5.6	14.2	29.2	6.9	Cu_12.81_Ni_1.62_Sb_4.11_S_8.47_Se_2.00_

^1^ Samples are identified according to the expected stoichiometric content of Ni and Se. ^2^ Compositions are calculated after adjusting atomic percentages to reflect a formula unit of 29 atoms (the number of atoms present in the standard tetrahedrite).

**Table 2 materials-16-00898-t002:** Tetrahedrite EDS semi-quantitative analysis on Cu_12−x_Ni_x_Sb_4_S_13−y_Se_y_ annealed samples.

Ni Content ^1^	Se Content ^1^	Cu (at%)	Ni (at%)	Sb (at%)	S (at%)	Se (at%)	Molecular Formula ^2^
0	0	42.0	0	14.4	43.6	0	Cu_12.18_Sb_4.18_S_12.64_
0.5	0	40.1	1.9	14.2	43.9	0	Cu_11.61_Ni_0.56_Sb_4.11_S_12.72_
0	0.5	40.6	0	14.1	44.1	1.2	Cu_11.78_Sb_4.08_S_12.79_Se_0.35_
0.5	0.5	40.2	1.4	13.9	43.5	1.3	Cu_11.67_Ni_0.41_Sb_4.03_S_12.62_Se_0.39_
1	0	39.0	2.6	13.8	44.7	0	Cu_11.30_Ni_0.75_Sb_4.0_S_12.95_
1	0.5	38.4	3.8	14.4	41.9	1.6	Cu_11.14_Ni_1.10_Sb_4.17_S_12.15_Se_0.45_
0	1	38.9	0	12.7	44.7	3.7	Cu_11.29_Sb_3.67_S_12.97_Se_1.08_
0.5	1	39.4	2.1	13.7	42.0	2.8	Cu_11.42_Ni_0.61_Sb_3.98_S_12.18_Se_0.81_
1	1	37.3	4.7	13.9	41.6	2.5	Cu_10.81_Ni_1.37_Sb_4.03_S_12.07_Se_0.72_
1.5	0	37.7	4.5	14.3	43.6	0	Cu_10.94_Ni_1.30_Sb_4.13_S_12.63_
1.5	0.5	38.3	4.0	14.1	42.5	1.2	Cu_11.12_Ni_1.15_Sb_4.07_S_12.32_Se_0.34_
1.5	1	37.3	4.8	14.2	41.2	2.6	Cu_10.83_Ni_1.38_Sb_4.12_S_11.93_Se_0.74_
0	1.5	42.5	0	14.4	39.2	3.9	Cu_12.33_Sb_4.16_S_11.37_Se_1.14_
0.5	1.5	39.8	1.4	14.7	40.5	3.6	Cu_11.55_Ni_0.40_Sb_4.250_S_11.74_Se_1.06_
1	1.5	38.4	3.6	14.0	40.1	3.9	Cu_11.13_Ni_1.05_Sb_4.05_S_11.63_Se_1.14_
1.5	1.5	37.5	4.5	13.9	40.3	3.8	Cu_10.86_Ni_1.3_Sb_4.04_S_11.68_Se_1.11_

^1^ Samples are identified according to the expected stoichiometric content of Ni and Se. ^2^ Compositions are calculated after adjusting atomic percentages to reflect a formula unit of 29 atoms (the number of atoms present in the standard tetrahedrite).

## Data Availability

The data presented in this study were measured experimentally or calculated numerically by the authors. Further details about the results may be requested to the authors.

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
