# Peer review of "Thermoelectric Properties of Nickel and Selenium Co-Doped Tetrahedrite"

_materials, 2023, doi:10.3390/ma16030898_

Round 1

Reviewer 1 Report

Cu12Sb4S13 is a promising thermoelectric material owing to its low cost, non-toxicity and especially the extremely low lattice thermal conductivity. This work reports the systematically investigation on the synthesis, phase identification, sample composition, thermoelectric properties of Ni and Se codoped Cu12Sb4S13. The research topic is quite interesting to the thermoelectric society, and the paper is well organized and presented. I would like to recommend the acceptance of this work for publication in Materials after a minor revision.

As the author wrote in the paper, Ni is one valence electron less than Cu, and thus the doping of Ni should increase the hole concentration of Cu12Sb4S13. Since Cu12Sb4S13 is a p-type semiconductor, the doping of Ni should increase the electrical conductivity or decrease the electrical resistivity of Cu12Sb4S13. However, the electrical conductivity of the Ni doped sample shows higher electrical resistivity than the pristine Cu12Sb4S13. Could the author explain this?

Author Response

Reviewer 1:

Cu12Sb4S13 is a promising thermoelectric material owing to its low cost, non-toxicity and especially the extremely low lattice thermal conductivity. This work reports the systematically investigation on the synthesis, phase identification, sample composition, thermoelectric properties of Ni and Se codoped Cu12Sb4S13. The research topic is quite interesting to the thermoelectric society, and the paper is well organized and presented. I would like to recommend the acceptance of this work for publication in Materials after a minor revision.

As the author wrote in the paper, Ni is one valence electron less than Cu, and thus the doping of Ni should increase the hole concentration of Cu12Sb4S13. Since Cu12Sb4S13 is a p-type semiconductor, the doping of Ni should increase the electrical conductivity or decrease the electrical resistivity of Cu12Sb4S13. However, the electrical conductivity of the Ni doped sample shows higher electrical resistivity than the pristine Cu12Sb4S13. Could the author explain this? 

R: We would like to thank Reviewer 1 for the valuable comments. Indeed, from a band structure point of view, doping with Ni is expected to increase the hole concentration and, consequently, the electrical conductivity. However, other aspects, like electron impurity scattering or grain boundary scattering, which can strongly influence the electrical conductivity behaviour, can also exist.  The manuscript was revised and updated in order to include such considerations.

Reviewer 2 Report

Thermoelectric materials have attracted intensive attention but are still limited by rare high-performance materials. In this manuscript, the authors studied the effects of doping with Ni and Se on the thermoelectric properties of tetrahedrite. The results are interesting and provide some suggestions for tuning thermoelectric materials. The manuscript can be accepted after some revisions.

1.      The reference should be provided in lines 116-117.

2.      The y-axis in figure 3d is not consistent with the caption. Moreover, Figure 8 a and b are mismatched with the caption.

3.      In lines 221-223, “with the highest value, achieving an estimated zT≈0.35 at 300 K, being observed for the Cu11.5Ni0.5Sb4S12.5Se0.5 composition.” However, we observe the zT≈0.3 in figure 3c. In addition, in lines 319-320, “It also points out the Cu11.5Ni0.5Sb4S12.5Se0.5 tetrahedrite as the one with the highest performance, presenting a zT around 0.30 at ~300K”, which is a mismatch with lines 221-223.

4.      Some writing mistakes need to be corrected. For example, in line 225, “κL=0.5 W K-1m-1”, and in line 229, “Cu12Sb4S12Se”.

5.      The format of the x-axis should be unified in Figures 4, 5, and 8. For example, the x-axis in Figure 4a is 10-75, but the abscissa in Figure 4b is 10-70.

6.      In lines 240-241, “All samples exhibit the peaks typical of the tetrahedrite phase (at 315 cm-1 and 348 cm-1)”, while it is hard to find the peak of 315 cm-1 in figure 5A. In addition, the author should explain why the peak of 348 cm-1 shifted and what caused the shift.

7.      In figure 9, When Ni=0, Se=1.5, the curve is missing 50-175 K.

8.      In lines 419-421, “A maximum PF of 1279.99 µ W/m.K2 at 300K is observed for this sample, followed by the Cu11NiSb4S12.5Se0.5 and Cu11.5Ni0.5Sb4S12Se compositions, with a PF=527.96 µ W/m.K2 and PF=335.56 µ W/m.K2 at 300 K, respectively”, which are mismatched with figure 10A. In addition, in lines 435-439, “With this approximation, a figure of merit zT=0.325 was calculated at 300K for the sample with Cu11.5Ni0.5Sb4S12.5Se0.5 composition (Figure 10 B), which is a particularly high value when compared to the reported results for tetrahedrites. [34,37,49,57] Other high zTs were also achieved with close compositions, zT= 0.207 for Cu11NiSb4S12.5Se0.5 and zT=0.118 for Cu11.5Ni0.5Sb4S12Se”, which are mismatched with figure 10B.

9.      The reason for higher thermoelectric performance by introducing Ni and Se should be provided in the conclusion.

Author Response

Thermoelectric materials have attracted intensive attention but are still limited by rare high-performance materials. In this manuscript, the authors studied the effects of doping with Ni and Se on the thermoelectric properties of tetrahedrite. The results are interesting and provide some suggestions for tuning thermoelectric materials. The manuscript can be accepted after some revisions.

R: We would like to thank Reviewer 2 for the valuable comments and suggestions. Below goes the detailed answer to such comments.

  1. The reference should be provided in lines 116-117.

R: References were provided.

  1. The y-axis in figure 3d is not consistent with the caption. Moreover, Figure 8 a and b are mismatched with the caption.

R: Figure 3d was remade and the y-axis is now consistent with the caption. The same was done for Figure 8 and no mismatch now exists.

  1. In lines 221-223, “with the highest value, achieving an estimated zT≈0.35 at 300 K, being observed for the Cu11.5Ni0.5Sb4S12.5Se0.5 composition.” However, we observe the zT≈0.3 in figure 3c. In addition, in lines 319-320, “It also points out the Cu11.5Ni0.5Sb4S12.5Se0.5 tetrahedrite as the one with the highest performance, presenting a zT around 0.30 at ~300K”, which is a mismatch with lines 221-223.

R: zT in line 222 was corrected to 0.30.

  1. Some writing mistakes need to be corrected. For example, in line 225, “κL=0.5 W K-1m-1”, and in line 229, “Cu12Sb4S12Se”.

R: Mistakes were corrected.

  1. The format of the x-axis should be unified in Figures 4, 5, and 8. For example, the x-axis in Figure 4a is 10-75, but the abscissa in Figure 4b is 10-70.

R: The format of the x-axis was unified in Figures 4, 5, and 8.

  1. In lines 240-241, “All samples exhibit the peaks typical of the tetrahedrite phase (at 315 cm-1 and 348 cm-1)”, while it is hard to find the peak of 315 cm-1 in figure 5A. In addition, the author should explain why the peak of 348 cm-1 shifted and what caused the shift.

R: We agree with the Reviewer 2 that in some samples it is hard to find the 315 cm-1 peak and just a small anomaly is observed. Moreover, a slight shift of the 348 cm-1 peak (up to a maximum 355 cm-1) is seen, which is related to changes in the energy of the molecular vibrations induced by substitutions. The manuscript was expanded and now includes these points.

  1. In figure 9, When Ni=0, Se=1.5, the curve is missing 50-175 K.

R: Due to its mechanical properties (it was too brittle), the low temperature electrical resistivity measurements were not possible for this sample. However, this do not influences the general conclusions of the manuscript.

  1. In lines 419-421, “A maximum PF of 1279.99 µ W/m.K2 at 300K is observed for this sample, followed by the Cu11NiSb4S12.5Se0.5 and Cu11.5Ni0.5Sb4S12Se compositions, with a PF=527.96 µ W/m.K2 and PF=335.56 µ W/m.K2 at 300 K, respectively”, which are mismatched with figure 10A. In addition, in lines 435-439, “With this approximation, a figure of merit zT=0.325 was calculated at 300K for the sample with Cu11.5Ni0.5Sb4S12.5Se0.5 composition (Figure 10 B), which is a particularly high value when compared to the reported results for tetrahedrites. [34,37,49,57] Other high zTs were also achieved with close compositions, zT= 0.207 for Cu11NiSb4S12.5Se0.5 and zT=0.118 for Cu11.5Ni0.5Sb4S12Se”, which are mismatched with figure 10B.

R: The values in the text and the Figure 10A and Figure 10B are correct, but probably the perspective induces wrong interpretations. Figure 10 was redrawn to be clear.

  1. The reason for higher thermoelectric performance by introducing Ni and Se should be provided in the conclusion.

R: The reason for higher thermoelectric performance by introducing Ni and Se was provided in the conclusion.
